# Serum Leucine-Rich Alpha-2 Glycoprotein in Quiescent Crohn’s Disease as a Potential Surrogate Marker for Small-Bowel Ulceration detected by Capsule Endoscopy

**DOI:** 10.3390/jcm11092494

**Published:** 2022-04-29

**Authors:** Teppei Omori, Yu Sasaki, Miki Koroku, Shun Murasugi, Maria Yonezawa, Shinichi Nakamura, Katsutoshi Tokushige

**Affiliations:** Institute of Gastroenterology, Tokyo Women’s Medical University, Tokyo 162-8666, Japan; sasaki.yu@twmu.ac.jp (Y.S.); koroku.miki@twmu.ac.jp (M.K.); murasugi.shun@twmu.ac.jp (S.M.); yonezawa.maria@twmu.ac.jp (M.Y.); nakamura.shinichi@twmu.ac.jp (S.N.); tokushige.ige@twmu.ac.jp (K.T.)

**Keywords:** small bowel capsule endoscopy, leucine-rich alpha-2 glycoprotein, Lewis score, quiescent, Crohn’s disease

## Abstract

Background: Small bowel (SB) lesions in quiescent Crohn’s disease (CD) are sometimes not identified by clinical activity or existing markers. We investigated the usefulness of a novel biomarker, leucine-rich α2-glycoprotein (LRG), for screening for the presence of SB ulcerative lesions detected by small-bowel capsule endoscopy (SBCE). Methods: We examined patients with a Crohn’s Disease Activity Index (CDAI) value < 150 and a C-reactive protein (CRP) value < 0.5 mg/dL with SB or SB colonic CD. The presence of small-bowel ulcerative lesions (≥0.5 cm) was grouped by SBCE results, and we then compared the groups’ LRG value to establish a cutoff value for screening for the presence of lesions. Results: In 40 patients with CD, the LRG values differed significantly between the patients with and without SB ulcerative lesions (Ul + 14.1 (2.1–16.5) μg/mL vs. Ul − 12.3 (9.3–13.5) μg/mL; *p* = 0.0105). The respective cutoff LRG values for the presence of SB ulcerative lesions was 14 μg/mL (areas under the ROC curve 0.77), with sensitivity 63.6%, specificity 82.8%, positive predictive values 58.3%, negative predictive values 85.7%, and accuracy 78%. Conclusion: These results indicate that LRG may be useful in predicting the presence of SB inflammation associated in patients with CD with CRP < 0.5 mg/dL and CDAI < 150, and in selecting patients for SBCE.

## 1. Introduction

Crohn’s disease (CD) is a chronic, destructive, progressive inflammatory disease that causes inflammation and intestinal damage mainly in the small and large bowel [1,2]. The small bowel in particular is involved in approximately 70% of cases of individuals with CD. It has been pointed out that small bowel lesions in CD are difficult to assess accurately with the Crohn’s Disease Activity Index (CDAI), which is a subjective assessment of disease activity in CD; it has also been posited that a CD patient’s C-reactive protein (CRP) value does not reflect the disease activity of small intestinal lesions [3].

In patients with CD, the Crohn’s Disease Endoscopic Index of Severity (CDEIS) and the simple endoscopic score for Crohn’s Disease (SES-CD) are generally determined by ileocolonoscopy before the use of endoscopy [4]. Although the CDEIS and SES-CD scores are correlated, the standard use of ileocolonoscopy does not adequately assess the deep small bowel on the mouth-side from the terminal ileum, and it may underestimate the presence of small bowel lesions [5,6].

Leucine-rich α2-glycoprotein (LRG) has attracted attention as a new biomarker in ulcerative colitis and CD [7,8,9,10]. LRG is an interleukin (IL)-6-independent 50-kD protein produced in the localized inflammation of the intestinal tract [7]. In CD, the identification of an LRG value ≥ 16 μg/mL (according to the manufacturer’s instructions; Sekisui Medical Co., Tokyo, Japan) was reported to be useful for discriminating patients in remission (CDAI < 150 and SES-CD < 4) from patients with active CD (CDAI ≥ 150 and SES-CD ≥ 4). LRG was also reported to be useful for discriminating disease activity even in CRP-negative CD [10]. These findings did not involve evaluation for deep small-bowel lesions on the mouth-side from the terminal ileum. However, extensive small bowel involvement has been shown to be a poor prognostic factor in CD [11].

Balloon-assisted enteroscopy (BAE) and small bowel capsule endoscopy (SBCE) have been used to visualize inflammation in the deep small bowel [12]. Studies of BAE conducted to identify small bowel lesions of CD indicated that small bowel ulcerative lesions ≥ 0.5 cm are an independent risk factor for relapse and surgery [3]. A study of SBCE reported that in patient CD with a CDAI < 150 or a CDAI < 220 that did not require new therapeutic intervention in 3 months, an SBCE score (i.e., Lewis score (LS)) ≥ 350 was predictive of relapse at 24 months [13]. The monitoring of small bowel lesions in patients with CD in clinical remission is of high clinical relevance. However, the use of BAE or SBCE as a monitoring tool in all patients is not feasible due to the invasiveness and cost of these procedures. If the presence of small bowel lesions can be predicted by LRG, a noninvasive biomarker, the use of LRG will contribute to the selection of patients who should undergo endoscopy. We conducted the present study to determine the association between LRG values and SBCE-diagnosed small-bowel active inflammatory lesions in patients with CD in clinical remission with a CDAI value < 150 and a CRP level < 0.5 mg/dL.

## 2. Materials and Methods

### 2.1. Patients Selection

We retrospectively evaluated the cases of 59 patients with small bowel or small bowel colorectal-type CD attending who had undergone: (1) a patency evaluation with the use of a patency capsule, (2) small bowel capsule endoscopy (SBCE), and (3) LRG measurement during the period July 2020 to December 2021 at Tokyo Women’s Medical University. We then excluded the patients who had a CDAI ≥ 150, a CRP value ≥ 0.5 mg/dL, with an interval between their SBCE and LRG measurements of >30  ±  7 days, colorectal CD, active colorectal ulcerative lesions ≥ 0.5 cm, active perianal lesions, stoma, age < 18 years, failure of patency capsule assessment, or failure of total small bowel observation. A final total of 40 patients was included in the analysis (Figure 1).

### 2.2. Evaluation Method

SBCE, which can image the entire small bowel, was used to identify small-bowel lesions. The serological markers CRP, LRG, and CDAI were measured within 30 ± 7 days of the SBCE. Inflammatory mucosal defects with white mucosa > 0.5 cm were defined as small-bowel ulcerative lesions [14], and their presence was confirmed by SBCE (Appendix A). The diameter of a typical small intestine is 2.5 cm, and we calculated the circumference of a quarter of a small intestine to be 2 cm. Based on that, we estimated the size of the ulcer. This ulcerative lesion was considered to be classified as a deep ulcer, a finding primarily associated with Crohn’s disease, according to the International Consensus Statement [15,16]. Each patient’s capsule endoscopic score, Lewis score (LS), Capsule Endoscopy Crohn’s Disease Activity Index (CECDAI) [4], and Crohn’s Disease Activity in Capsule Endoscopy (CDACE) were also calculated [17].

The primary endpoint of the study was to compare LRG values according to the presence or absence of small intestinal ulcerative lesions. We also compared whether LRG values differed in LS ≥ 350. As a result, we determined the appropriate LRG screening value by setting the cutoff value using the Youden index from the receiver operating characteristic (ROC) curve of the LRG values measured in the presence of small intestinal ulcerative lesions and an LS ≥ 350. We then calculated the sensitivity, specificity, positive predictive value (PPV), negative predictive value (NPV), and accuracy of the cutoff value. As a secondary endpoint, we calculated the correlations between the CRP values, LRG values, and each SBCE score, and we compared the background factors and SBCE scores of the patients with quiescent CD grouped by the previously reported LRG cutoff value (16 μg/mL) and our newly calculated cutoff value. Finally, LRG values and SBCE scores in the use of biological agents were evaluated.

### 2.3. SBCE Procedure

Intestinal patency was confirmed in all patients before the SBCE procedure by the use of patency capsules (PC; Medtronic, Minneapolis, MN, USA). SBCE was performed by practitioners with experience conducting > 1200 SBCE procedures. The capsule endoscopy device used for all patients was the PillCam™ SB3 (Medtronic, Minneapolis, MN, USA). The patient was instructed to fast from 21:00 on the day before the SBCE examination. At 8:00 the following morning, oral mosapride (15 mg) was administered as a pretreatment drug. An intestinal cleansing procedure was considered unnecessary for the SBCE. The timepoint at which the patient swallowed the SBCE with water marked the start of the procedure. Drinking water was provided 2 h after the SBCE ingestion, and a meal was provided 4 h after the patient swallowed the SBCE endoscope. The excretion of the capsule was confirmed visually after the completion of the examination.

### 2.4. Ethical Considerations

The study protocol was approved by the Institutional Ethics Review Committee of Tokyo Women’s Medical University, and each patient provided written informed consent (IRB no. 2021-0114).

### 2.5. Statistical Analysis

All the data are expressed as the median (interquartile range (IQR)). Wilcoxon’s test was used in the univariate analysis of background factors. Spearman’s rank correlation coefficient was used to analyze the correlations of the LRG score with the CDACE, LS, CECDAI, and CDAI scores. Probability (*p*)-values < 0.05 were considered significant. With regard to the presence of small intestinal ulcers (≥0.5 cm) and LS ≥ 350, we calculated the cutoff value of each score using the Youden index from the ROC curve and computed the sensitivity, specificity, PPV, NPV, and accuracy. The JMP statistical analysis software (ver. 12; SAS, Cary, NC, USA) was used in all analyses.

## 3. Results

### 3.1. Patient Characteristics

The background of the 40 patients was as follows (Table 1): Age 36.4 (32–50.6) years; 28 males/12 females (70%/30%), disease duration 12.8 (7.2–17.6) years, 20 patients (50%) with previous intestinal surgery, 10 patients (25%) with previous anorectal lesions, 13 patients with small bowel-type CD (32.5%), and 27 patients with small bowel colon-type CD (67.5%). Among the patients with small bowel colon-type CD, the colorectal evaluation was available in nine (29.6%) patients at the same time as the SBCE (interval: 20.5 (4.8–35.8) days), and the measurement interval between SBCE and LRG was 6 (0–11) days. 

### 3.2. Summary of Capsule Endoscopy Findings

As shown in Table 2, the patients’ median (IQR) scores calculated based on their SBCE results were LS: 0 (0–192), CECDAI: 3 (0–6), and CDACE: 211 (0–420). Among the 40 patients, 26 (65%) had an LS < 135, 10 (25%) had an LS ≥ 135 to <790, and 4 (10%) had an LS ≥ 790. Six patients (15%) had a stenotic lesion through which the SBCE passed. Seven patients (17.1%) had inflammatory findings in the first and/or second segment of the LS.

### 3.3. Evaluation Using LRG for Small-Bowel Ulcerative Lesions

Based on their SBCE results, 11 of the 40 patients (27.5%) had small-bowel ulcerative lesions with a diameter ≥ 0.5 cm. The LRG values were significantly different between the groups of patients with and without the presence of small-bowel ulcerative lesions (ulcer (Ul) + 14.1 (12.1–16.5) μg/mL vs. Ul − 11.7 (9.3–13.5) μg/mL, *p* = 0.0105). However, there was no significant difference in CRP levels between the patients with and without small intestinal ulcerative lesions (Ul + 0.09 (0.06–0.18) mg/dL vs. Ul − 0.08 (0.05–0.13) mg/dL, *p* = 0.3018) (Appendix A).

The LRG value with the highest area under the curve (AUC) was calculated from the ROC curves obtained in the presence and absence of small intestinal ulcerative lesions: LRG 14 μg/mL (AUC 0.77, Figure 2). The use of LRG 14 μg/mL to detect the presence of small intestinal ulcerative lesions showed 63.6% sensitivity, 82.8% specificity, 58.3% PPV, 85.7% NPV, and 78% accuracy.

### 3.4. Evaluation Using LRG for Lewis Scores ≥ 350

Based on their SBCE results, five of the forty patients (12.5%) had an LS ≥ 350. The LRG values were significantly different between the groups of patients with an LS ≥ 350 (15.3 (14.4–17.3) μg/mL) and those with an LS < 350 (11.7 (9.7–13.6) μg/mL), *p* = 0.0030 (Appendix A). From the ROC curves obtained with LS ≥ 350, the LRG value with the highest AUC was LRG 14 μg/mL (AUC 0.92, Figure 3). The detection ability of LRG 14 μg/mL with LS ≥ 350 was as follows: 100% sensitivity, 80% specificity, 41.7% PPV, 100% NPV, and 82.5% accuracy.

### 3.5. Comparison of Patients with LRG Values

When we divided the patients into two groups based on the cutoff LRG 14 μg/mL (i.e., <14 μg/mL and ≥14 μg/mL) and compared the groups, we observed significant differences in the LS, CECDAI, and CDACE scores in both groups as follows: LS: 268 (45–3272) vs. 0 (0–0), *p* = 0.0002; CECDAI: 7.5 (3.5–12) vs. 3 (0–5.8), *p* = 0.0027; CDACE: 421 (312–913) vs. 210 (0–310), *p* = 0.0031. There was also a significant difference in small bowel ulcerative lesions and LS ≥ 350 between the two groups (*p* = 0.0078, 0.0012, respectively) (Table 3).

When the patients were grouped into two groups based on the existing cut-off value of LRG 16 μg/mL (i.e., <16 μg/mL and ≥16 μg/mL), the respective SBCE score values (LS, CECDAI, and CDACE) were significantly different in both groups (Appendix A). The presence of ≥0.5 cm ulcerative lesions and LS ≥ 350 could not be determined by using the cutoff LRG 16 μg/mL, whereas the use of LRG 14 μg/mL as the cutoff could determine it (Table 3).

### 3.6. The Correlations between SBCE Score and the LRG Values

The correlations between each SBCE score and the LRG values were as follows: LS: Spearman’s rank correlation coefficient (ρ) = 0.5832, *p* < 0.0001; CECDAI: ρ = 0.5985, *p* < 0.0001; and CDACE: ρ = 0.5495, *p* = 0.0002. The correlations between each SBCE score and the CRP values were as follows: LS: ρ = 0.1540, *p* = 0.3428; CECDAI: ρ = 0.2139, *p* = 0.1851; and CDACE: ρ = 0.2074, *p* = 0.1991. The correlations between each SBCE score and CDAI were as follows: LS: ρ = 0.1363, *p* = 0.4017; CECDAI: ρ = 0.3116, *p* = 0.0503; and CDACE: ρ = 0.2179, *p* = 0.1767. There were thus significant correlations between the SBCE scores and LRG. There was no correlation between the CRP and CDAI values and each SBCE score, as shown in Table 4.

### 3.7. Influence of Biologics on LRG Values and SBCE Scores

The LRG values according to the use of biological agents (TNFα inhibitor *n* = 20, anti-integrin inhibitor *n* = 1, anti-IL12/23p40 inhibitor *n* = 4) were significantly different: LRG 13.9 (11.8–14.4) vs. 10.8 (9.5–13.3) without biological agent vs. with biological agents (*p* = 0.0314). In addition, the respective SBCE scores for both groups were LS: 0 (0–225) vs. 0 (0–135) (*p* = 0.6071), CECDAI: 6 (3–10) vs. 3 (0–4) (*p* = 0.0224), CDACE: 420 (210–730) vs. 210 (0–311) (*p* = 0.0357). The SBCE scores were also significantly different between patients with and without biological agents (Appendix A).

## 4. Discussion

While it has been reported that the presence of small-bowel lesions in CD cannot be determined by clinical activity or existing markers, small bowel ulcerative lesions ≥ 0.5 cm are an independent risk factor for relapse and surgery [3], and the monitoring of such lesions is of high clinical significance even in patients with CD who are judged to be in clinical remission. We therefore investigated whether the use of a patient’s LRG value could predict the presence or absence of small bowel ulcerative lesions in quiescent patients with CD in clinical remission with a CRP level < 0.5 mg/dL. The results showed that a lower LRG value of 14 μg/mL rather than the previously reported LRG cutoff value of 16 μg/mL significantly predicted the presence of small intestinal ulcerative lesions ≥ 0.5 cm in diameter.

The prediction of ≥0.5 cm small bowel ulcerative lesions by LRG values was reported in a study of BAE [14]. In that report, the detection sensitivity was 79%, with 82% specificity, 93% PPV, and 58% NPV at the cutoff value of 13.4 μg/mL LRG. Another report indicated that the LRG value predicted a modified SES-CD score of 0, which represents only the inflammatory findings of SES-CD [9]. In that report, AUC 0.82 and 77% accuracy were obtained at a cutoff LRG value of 13 μg/mL, indicating that the prediction ability of LRG was equivalent to that of calprotectin. The discrepancy between the findings obtained in those studies and our present investigation may be explained by the fact that our patient series was limited to those with quiescent CD with clinical remission and negative CRP values, and the monitoring modalities used were different. The use of SBCE, which can image the entire small bowel, may have increased the visualized bowel length and contributed to the identification of lesions, resulting in different cutoff values.

Although the relationship between SBCE scores and relapse risk is still unclear, it was reported that an LS ≥ 350 is useful for predicting relapse after 24 months post baseline [13]. In the present study, the use of the LRG value < 14 μg/mL was able to significantly discriminate patients with an LS < 350 with 80% specificity and the NPV of 100%. This suggests that the use of LRG may be able to select patients with inflammatory findings—even those in clinical remission—who should be prioritized for a small bowel search. Our analyses also revealed that LRG was correlated with each SBCE score (LS, CECDAI, and CDACE) in patients with quiescent CD. Thus, LRG may reflect the extent of small bowel lesions detectable by SBCE that are not reflected by the CDAI or the CRP level.

In addition, biologic use status was associated with significantly lower LRG values and significantly lower capsule endoscopy scores in cases with CDAI < 150 and negative CRP. On the other hand, LS ≥ 350 and the presence of small intestinal ulcerative lesions larger than 0.5 cm did not differ significantly. It is necessary to be careful in interpreting these results, as the fact of using biologics may be influenced by the high potential disease activity, among other factors. However, it may be suggested that LRG measurement may be one of the indicators to evaluate therapeutic intervention in small intestinal lesions as well.

The strength of this study is that it is the first to compare LRG values of the entire small bowel in patients with CD with the use of SBCE, a modality that enables the easy observation of the entire small bowel; however, SBCE is expensive, and its indications must be carefully considered. It may be possible to predict patients with active small-bowel lesion CD who need to be evaluated by SBCE by measuring the LRG value beforehand.

There are some study limitations to consider. The cutoff value for LRG may be different in patients with small-bowel stenosis that cannot be passed by SBCE. The study’s design was retrospective and included a small number of patients (*n* = 40) at a single institution. Calprotectin was not measured; this is because the measurement of calprotectin in patients with CD is not covered by Japan’s health insurance system. However, a comparison of the efficacy of LRG and calprotectin has been reported [9], and further investigation of this parameter is warranted. No bowel cleansing agents were used in preparation for SBCE in this study. Therefore, there could potentially be lesions that were not visualized by capsule endoscopy. In addition, since we included patients with small-bowel colonic CD and since only 30% of the patients underwent colonoscopy at the same time that their SBCE was conducted and their LRG was measured, the activity of the colonic lesions may have affected the LRG values. However, the CDAI and the CRP level have been shown to be poor reflectors of the activity of small-bowel lesions [18,19,20], and thus, active lesions of the colon may be more reflective of clinical activity. All of the patients included in this study had CDAI values < 150 and CRP levels < 0.5 mg/dL, which we believe reduced this effect.

## 5. Conclusions

We suggest that the determination of the LRG value may be useful for (1) predicting the presence of small bowel inflammation associated with relapse in patients with CD with CRP values < 0.5 mg/dL and a CDAI < 150, and (2) selecting patients for SBCE. However, a multicenter validation in a larger number of patients is needed to strengthen this conclusion.

## Figures and Tables

**Figure 1 jcm-11-02494-f001:**
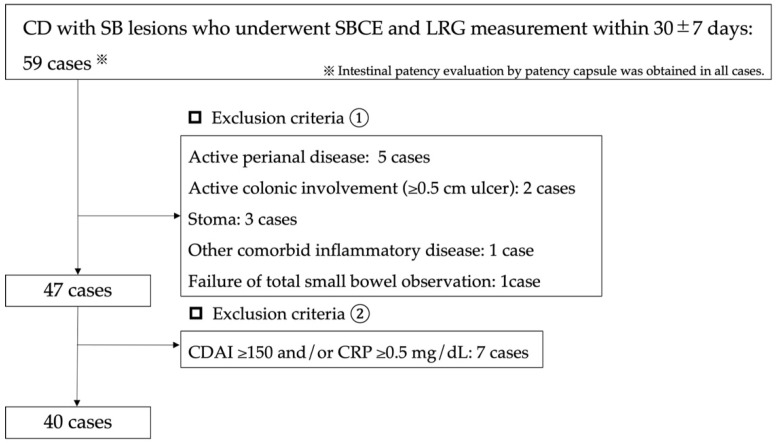
Study profile.

**Figure 2 jcm-11-02494-f002:**
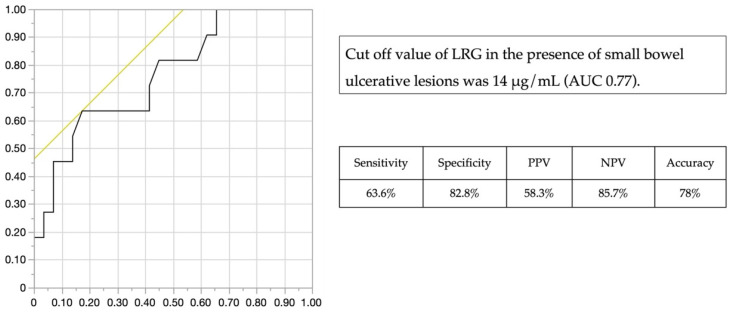
ROC curve of LRG for small bowel ulcerative lesion (≥0.5 cm). AUC; Area under the curve, LRG; Leucine-Rich Alpha-2 Glycoprotein, NPV: negative predictive value, PPV: positive predictive value, ROC; Receiver operating characteristic.

**Figure 3 jcm-11-02494-f003:**
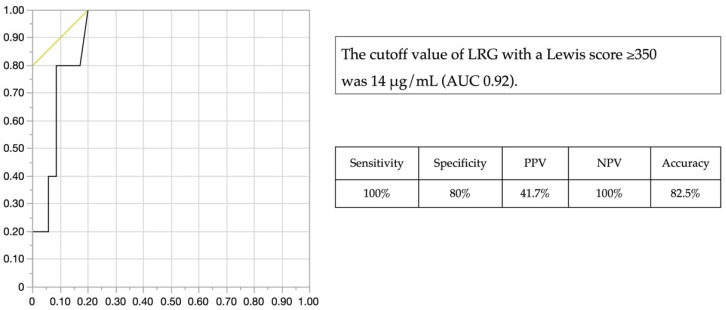
ROC curve of LRG for LS ≥ 350. AUC; Area under the curve, LRG; Leucine-Rich Alpha-2 Glycoprotein, LS; Lewis score, NPV: negative predictive value, PPV: positive predictive value, ROC; Receiver operating characteristic.

**Table 1 jcm-11-02494-t001:** Demographic and clinical characteristics of the patients with CD.

	*n* = 40 (%), Median (IQR)
Males	28 (70)
Age, years	36.4 (32–50.6)
Disease duration, years	12.8 (7.2–17.6)
Montreal classification:	
A1/A2/A3	5 (12.5)/30 (75)/5 (12.5)
B1/B2/B3	18 (45)/12 (30)/10 (25)
L1/L2/L3	13 (32.5)/0 (0)/27 (67.5)
Evaluation for L3 by colonoscopy within 2 months of their SBCE, *n*, days	8 (29.6), 20.5 (4.8–35.8)
Active colonic involvement	0 (0)
Perianal disease	10 (25)
Active perianal disease involvement	0 (0)
Past intestinal surgery	20 (50)
Smoking non/current/past	26 (65)/9 (22.5)/5 (12.5)
Medication	
5ASA	33 (82.5)
Elemental diet	19 (47.5)
PSL	1 (2.5)
AZA	8 (20)
Anti-TNFα inhibitor	20 (50)
IL-12/23p40 inhibitor	4 (10)
Integrin inhibitor	1 (2.5)
WBC/μL	5800 (4620–7488)
Hb, g/dL	13.7 (12.2–14.7)
Ht, %	40.7 (37–44)
Plt × 10^4^/μL	24.6 (21.1–29.2)
Alb, g/dL	4.4 (4.2–4.7)
CRP, mg/dL	0.08 (0.05–0.15)
LRG, μg/mL	12.3 (8.9–14.1)
CDAI	67 (29.5–89.8)
Interval between SBCE and CDAI/biomarker evaluation, days	6 (0–11)

The data are median (interquartile range). 5ASA: 5-aminosalicylic acid, Alb: albumin, AZA: Azathioprine, CDAI: Crohn’s Disease Activity Index, CDACE: Crohn’s Disease Activity in Capsule Endoscopy, CECDAI: Capsule Endoscopy Crohn’s Disease Activity Index, CRP: C-reactive protein, Hb: hemoglobin, Ht: hematocrit, IQR: interquartile range, LRG; Leucine-Rich Alpha-2 Glycoprotein, SBCE: small-bowel capsule endoscopy, WBC; White blood cell.

**Table 2 jcm-11-02494-t002:** Capsule Endoscopy score and findings.

		Total *n* = 40
SBCE score	Lewis score	0 (0–192)
	CECDAI	3 (0–6)
	CDACE	211 (0–420)
SB Ulcer ≥ 0.5 cm (%)		11 (27.5)
Lewis score ≥ 350 (%)		5 (12.5)
Inflammation on 1st and/or 2nd tertile in Lewis score (%)		7 (17.5)
Lewis score < 135, ≥135–<790, ≥790 (%)		26 (65)/10 (25)/4 (10)
No inflammation (Lewis score = 0) (%)		25 (62.5)
Passable stenosis (%)		6 (15)

The data are median (interquartile range). Abbreviations are explained in the footnote of Table 1.

**Table 3 jcm-11-02494-t003:** Comparison of patients with LRG < vs. ≥14 μg/mL.

		Total	LRG ≥ 14 μg/mL	LRG < 14 μg/mL	*p*-Value
		*n* = 40	*n* = 12	*n* = 28	
SBCE score	Lewis score	0 (0–192)	268 (45–3272)	0 (0–0)	0.0002
	CECDAI	3 (0–6)	7.5 (3.5–12)	3 (0–5.8)	0.0027
	CDACE	211 (0–420)	421 (312–913)	210 (0–310)	0.0031
Presence of small-bowel ulcer, ≥0.5 cm (%)		11 (27.5)	7(58.3)	4 (14.3)	0.0078
Lewis score ≥ 350 (%)		5 (12.5)	5 (41.7)	0 (0)	0.0012
Biomarkers/clinical activity:					
Hb, g/dL		13.7 (12.2–14.7)	14.2 (12.7–14.7)	13.6 (12.1–15)	0.4339
Plt × 10^4^/μL		24.6 (21.1–29.2)	24.9 (23.7–30)	23.6 (21–29)	0.2747
Alb, g/dL		4.4 (4.2–4.7)	4.3 (4.1–4.4)	4.5 (4.2–4.8)	0.0218
CRP, mg/dL		0.08 (0.05–0.15)	0.12 (0.08–0.33)	0.08 (0.05–0.11)	0.0275
LRG, μg/mL		12.3 (8.9–14.1)	15 (14.2–16.7)	10.3 (9.2–12.9)	<0.0001
CDAI		67 (29.5–89.8)	83.5 (63.5–109)	64 (28–86)	0.0550

The data are median (interquartile range). Abbreviations are explained in the footnote of Table 1.

**Table 4 jcm-11-02494-t004:** Correlation of SBCE scores and the values of LRG, CRP, and the CDAI.

	Spearman’s Rank Correlation Coefficient (ρ)	*p*-Value
	vs. LRG	
CRP	0.3707	0.0186
CDAI	0.4768	0.0019
Lewis score	0.5832	<0.0001
CECDAI	0.5985	<0.0001
CDACE	0.5495	0.0002
	vs. CRP	
CDAI	0.0693	0.6710
Lewis score	0.1540	0.3428
CECDAI	0.2139	0.1851
CDACE	0.2074	0.1991
	vs. CDAI	
Lewis Score	0.1363	0.4017
CECDAI	0.3116	0.0503
CDACE	0.2179	0.1767

Abbreviations are explained in the footnote of Table 1.

## Data Availability

The data presented in this study are available on request from the corresponding author.

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
