# Peer review of "Serum Leucine-Rich Alpha-2 Glycoprotein in Quiescent Crohn’s Disease as a Potential Surrogate Marker for Small-Bowel Ulceration detected by Capsule Endoscopy"

_jcm, 2022, doi:10.3390/jcm11092494_

Round 1
Reviewer 1 Report
I suggest to :
- modify the presentation of the aim, you should follow the PICO statement.
- About the methods, the study design and patients enrolment should be clarified.
- The tables must be improved to make it easier to read.
Author Response
Reviewer 1
I suggest to :
modify the presentation of the aim, you should follow the PICO statement.
About the methods, the study design and patients enrolment should be clarified.
The tables must be improved to make it easier to read.
Response: We thank the Reviewer for their comment. In this manuscript, P = CRP/CDAI negative CD patients with small bowel lesions who underwent SBCE, E = with SB ulcer, C = without SB ulcer, O = difference in LRG value. The method was modified accordingly. In addition, the study design was clearly stated to be a retrospective study (Line 69, Line 102-107). In Furthermore, the presentation of all tables was improved.
Reviewer 2 Report
This is a useful paper, with some interesting conclusions, but it needs attention on a few points:
one is the language and the overall presentation. If not already done, the authors are advised a) to reread their paper, and b) to consult a native speaker.
The other is the very thin reference list, excluding relevant works in the past and essential nomenclature consensus publications. This should be addressed, and the following papers should be referenced:
https://pubmed.ncbi.nlm.nih.gov/32213061/
https://pubmed.ncbi.nlm.nih.gov/33662785/
Author Response
Reviewer 2
This is a useful paper, with some interesting conclusions, but it needs attention on a few points:
one is the language and the overall presentation. If not already done, the authors are advised a) to reread their paper, and b) to consult a native speaker.
The other is the very thin reference list, excluding relevant works in the past and essential nomenclature consensus publications. This should be addressed, and the following papers should be referenced:
https://pubmed.ncbi.nlm.nih.gov/32213061/
https://pubmed.ncbi.nlm.nih.gov/33662785/
Response: We thank the Reviewer for the comment. We have reread the entire document and asked the MPDI-recommended proofreader to make the corrections. We also thank you for presenting an important manuscript on consensus statements regarding lesions. It is important for describing ulcerative lesions in this manuscript and has been added (Line 97-99).
Reviewer 3 Report
The article is useful regarding the necessity of using SBCE for patients with CD. In my experience is better to use PEG or another method for intestinal cleaning for enhancing the results.
Regarding the study, I also suggest a correlation between the treatment received by the patients and the presence of SB lesions and the value of LRG.
Author Response
Reviewer 3
The article is useful regarding the necessity of using SBCE for patients with CD. In my experience is better to use PEG or another method for intestinal cleaning for enhancing the results.
Regarding the study, I also suggest a correlation between the treatment received by the patients and the presence of SB lesions and the value of LRG.
Response: We thank the Reviewer for the comment. Regarding the pretreatment before capsule endoscopy, it is an important point and we have added it to the Limitations section (Line 251-253).
We also appreciate your suggestion regarding the relationship between treatment and LRG and SB lesions. We performed an analysis in this regard and found that patients with CDAI <150 and CRP <0.5 but on biologic agents had lower LRG and milder SBCE scores. This result is newly added (Line 195-201, Line 233-239, Suppl. Table S4).
Round 2
Reviewer 3 Report
The manuscript is better now.
Author Response
Dear Reviewer.
Thank you for your peer review.